# GCML: Grounding Complex Motions using Large Language Model in 3D Scenes

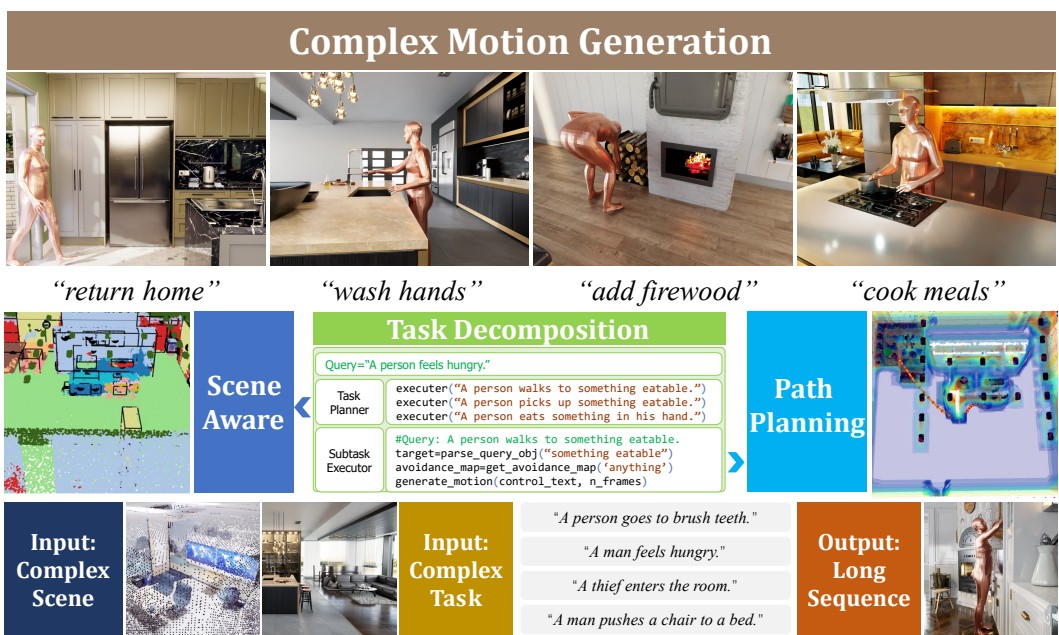

Figure 1: Different from previous methods that primarily focus on generating simple motions, our approach is designed to handle more complex actions. It leverages a Large Language Model for human action inference, task decomposition, and path planning. In combination with a 3D Visual Grounding Model for scene perception, this enables the generation of intricate, extended motions with complex scene and textual input.

## Abstract

To solve the problem of generating complex motions, we introduce GCML (Grounding Complex Motions using a Large Language Model). This method supports complex texts and scenes as inputs, such as mopping the floor in a cluttered room. Such everyday actions are challenging for current motion generation models for two main reasons. First, such complex actions are rarely found in existing HSI datasets, which places high demands on the generalization capabilities of current data-driven models. Second, these actions are composed of multiple stages, with considerable variation, making it difficult for models to understand and generate the appropriate motions. Current methods in the HSI field can control the generation of simple actions under multiple constraints, such as walking joyfully toward a door, but they cannot handle the complexity of tasks like the one described above. By incorporating a Large Language Model and a 3D Visual Grounding Model into the HSI domain, our approach can decompose a complex user prompt into a sequence of simpler subtasks and identify interaction targets and obstacles within the scene. Based on these subtask descriptions and spatial control information, the Motion Generation Model generates a sequence of full-body motions, which are then combined into a long motion sequence that aligns with both the user's input and the scene semantics. Experimental results

demonstrate that our method achieves competitive performance for simple action generation on the HUMANISE dataset and the generalization evaluation set. For complex motion generation, we created a new evaluation set by automatically generating possible behaviors of virtual humans in common indoor scenes, where our method significantly outperforms existing approaches. Project Page: https://anonymous.4open.science/w/GCML-4562/

# 1 INTRODUCTION

Research within Human-Scene Interaction (HSI) has advanced considerably in terms of modeling the sophisticated interaction between people and their environments (Zhao et al., 2023); (Zhang et al., 2022). It has also succeeded in synthesizing high-fidelity human motion in various scenes (Hassan et al., 2021); (Zhang & Tang, 2022); (Wang et al., 2022a). However, two major challenges exist:

**(1) Generative models depend heavily on large amounts of high-quality paired data.** The collection of HSI datasets and their annotations is time-consuming and labor-intensive task. Although many efforts have been made (Hassan et al., 2019); (Wang et al., 2022b);(Jiang et al., 2024), challenges like limited diversity in actions, overly simplistic descriptions, and inadequate dataset sizes still exist. On the other hand, existing datasets that only contain human motion, like AMASS (Mahmood et al., 2019) and HumanML3D (Plappert et al., 2016); (Guo et al., 2022b), are quite substantial in terms of data scale and motion description. Diffusion-based methods trained on these datasets are capable of generating high-quality human motions (Tevet et al., 2023) based on textual descriptions. Recent works (Karunratanakul et al., 2023); (Xie et al., 2023) also enable users to manipulate the style of generated actions via textual prompts and enhance precision using spatial constraints. Inspired by this, we propose tackling scene awareness and motion generation separately. This strategy helps mitigate the shortage of high-quality paired HSI data by maximizing the use of existing human motion and 3D visual grounding datasets.

**(2) Only Common and simple motions can be generated.** Due to the lack of diversity in HSI datasets, the majority of current research that generates human motions in scenes tends to focus on producing common and simple movements such as walking, lying, and sitting (Wang et al., 2022b); (Jiang et al., 2024); (Wang et al., 2024). However, industries with a need for human motion generation often require more than these elementary motions. For instance, video game characters should navigate to specified locations and interact with particular targets, while animated movie characters need to perform everyday tasks such as brushing their teeth, cooking, or watering plants. Such complex motions are rare in paired datasets, thus current data-driven methods for human motion generation typically fail to produce them. However, complex motions are often composed of simpler ones, and a practical solution to this challenge is to utilize Large Language Models for task decomposition and reasoning (Huang et al., 2023b); (Lin et al., 2024).

To alleviate these limitations, we propose a novel method called Grounding Complex Motions using Large Language Models (GCML), depicted in Figure 1. Our method can generate complex actions like washing hands or cooking meals, accommodating complex scene and textual input as well as producing human motions of long sequences. Furthermore, by incorporating Large Language Models and 3D Visual Grounding Models, our approach creatively generates motions consistent with textual descriptions based on the objects and their layout within a scene. For example, a hungry person in a scene, upon not finding available food, will attempt to open a refrigerator in search of food.

Our method operates by taking both text and scene inputs. The text is first processed by the Task Planner, which breaks it down into multiple sub-tasks, each corresponding to a simple motion sequence. Next, the Sub-task Executor identifies the target objects within the scene for interaction and generates control descriptions for each sub-task with the help of the 3D Visual Grounding Model. These control descriptions and spatial data are sent to the Motion Generation Model, which refines them into whole-body motion sequences for each sub-task. Finally, these sequences are combined into a complete, unified motion.

We have thoroughly evaluated GCML on a variety of benchmarks. Experimental results on the HUMANISE dataset reveal that even for simple tasks, without relying on HUMANISE training data, the

performance of our method is comparable to the current state-of-the-art data-driven approaches. On the generalization evaluation set proposed by (Wang et al., 2024), our method outperforms afford-motion across most metrics. Additionally, to specifically evaluate GCML's performance in complex motion generation, we introduced a new evaluation set, the Complex Motion Evaluation Set. Our method was the only one capable of producing satisfactory outcomes on this set.

Our key contributions are as follows:

- We introduce a new task along with a corresponding evaluation set: Complex Motion Generation. The gaming and animation industries increasingly require not just simple actions like walking or sitting, but also the ability to think and interact with surrounding objects like a human. We hope this new task will inspire the development of more methods for generating realistic virtual humans.

- As an attempt to address the challenge of complex human motion generation, we present the GCML framework. By integrating a Large Language Model and a 3D Visual Grounding Model into the HSI domain, our approach circumvents the lack of high-quality paired data, allowing for the generation of motions like brushing teeth and watering plants, which previous methods could not achieve.

- We validated the effectiveness of our proposed method across three datasets. GCML achieves comparable performance in simple motion generation, while in complex motion generation, it consistently outperforms others across all metrics.

## 2 RELATED WORK

### 2.1 CONDITIONAL HUMAN MOTION GENERATION

Human-Scene Interaction can be seen as the generation of human motions conditioned on both language descriptions and scene context. A wide range of conditions have been explored for controlled motion generation, such as past motions (Yuan & Kitani, 2020); (Cao et al., 2020); (Xie et al., 2021), music (Tseng et al., 2023); (Li et al., 2021), text (Petrovich et al., 2023); (Guo et al., 2022c); (Tevet et al., 2022); (Chen et al., 2023); (Kim et al., 2023), objects (Ghosh et al., 2023); (Kulkarni et al., 2024); (Xu et al., 2023), and scenes (Huang et al., 2023a); (Wang et al., 2022b);. Furthermore, recent work has added spatial constraints to text-guided motion generation (Text2Motion). MDM (Tevet et al., 2023) and priorMDM (Shafir et al., 2023) use motion inbetweening during the diffusion denoising process to replace key control frames, allowing for human motion generation that fits spatial constraints without sacrificing motion quality. GMD (Karunratanakul et al., 2023) introduces a two-stage motion diffusion model to handle sparse control signals, reducing jitter in controlled frames. Omnicontrol (Xie et al., 2023), employing a ControlNet-inspired approach, applies spatial constraints during motion generation and enables control over any key joint.

In the domain of Human Scene Interaction, language, and scene context are the two primary conditioning factors. HUMANISE (Wang et al., 2022b) introduced a comprehensive dataset and developed a cVAE-based model capable of generating motions that respond to both textual descriptions and scene interactions. TRUMANS (Jiang et al., 2024) proposed an even larger human-scene interaction dataset and used autoregressive conditional diffusion to generate HSI motions of any length. AffordMotion (Wang et al., 2024) employed scene affordances as intermediate representations, combining scene embeddings and language-guided motion generation in a two-stage approach. However, these methods are limited by their dependence on training data, generating only the common motion types seen in the datasets. Our work resolves this by separating scene understanding from motion generation, thus mitigating the reliance on specific datasets.

### 2.2 LARGE LANGUAGE MODELS FOR HUMAN MOTION GENERATION

Large Language Models have seen extensive research in the fields of intelligent agents (Wang et al., 2023); (Lin et al., 2023); (Park et al., 2023) and robotics (Blukis et al., 2020); (Yang et al., 2024); (Liu et al., 2024). The task planning and commonsense reasoning capabilities of pre-trained language models have substantially improved the ability of embodied agents to interpret their environment and tackle complex tasks in these domains.

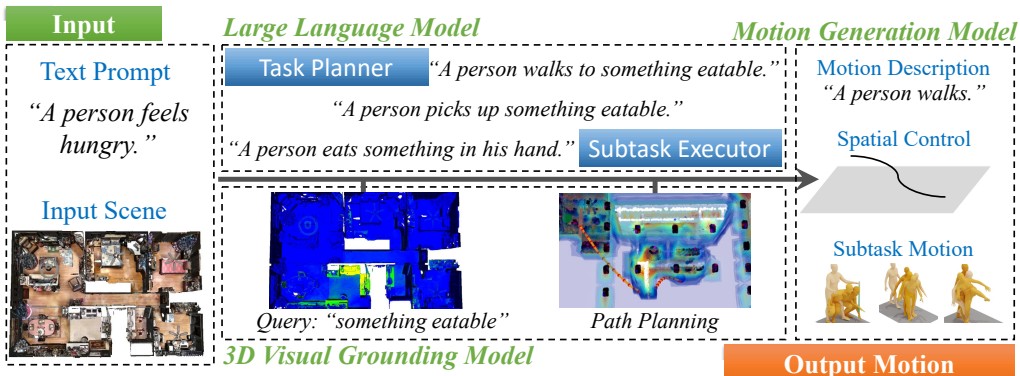

Figure 2: Overview of our method. To generate complex motions that align with text descriptions and scene semantics, GCML first utilizes a Large Language Model to break the task down into a sequence of simpler subtasks. Then, it uses the 3D Visual Grounding Model to extract scene information and generate control data for certain human joints. Finally, the Motion Generation Model produces the full-body motion frames.

In Human Motion Generation, MotionGPT (Jiang et al., 2023) introduced a general motion generator capable of embedding multimodal signals as special input tokens in Large Language Models, allowing it to handle various tasks related to human motion recognition and generation. UniHSI (Xiao et al., 2023) leveraged Large Language Models to decompose language descriptions into a contact chain—a sequence of interactions between human joints and object parts—enabling the continuous generation of motions interacting with multiple objects.

However, both approaches have limitations. MotionGPT cannot perceive the scene, while UniHSI requires fine-grained annotation of scene objects to generate accurate interaction motions. Additionally, UniHSI is unable to generate motions that do not involve contact with fixed objects, such as brushing teeth or eating, limiting its applicability.

### 2.3 3D VISUAL GROUNDING

3D scene understanding has been extensively researched, particularly in the fields of vision and robotics. Core tasks include 3D object classification (Wu et al., 2015) , 3D object detection and localization (Caesar et al., 2020); (Chen et al., 2020), 3D semantic and instance segmentation (Behley et al., 2019); (Liao et al., 2022), and 3D affordance prediction (Deng et al., 2021). A recent approach, OpenScene (Peng et al., 2023), extracts CLIP features for each 3D point, supporting open-vocabulary queries and segmentation based on any text input, making it a versatile tool for scene understanding in our work.

In Human Scene Interaction tasks, few approaches utilize existing 3D scene understanding models to assist in generating motions that align with scene semantics. Instead, data-driven methods typically train models by mapping specific 3D structures to actions, but this introduces two major limitations: the reliance on the quantity and quality of paired data limits performance, and biases in the data may restrict certain interactions. For example, interactions with chairs are typically confined to sitting, ignoring other potential uses. Utilizing existing 3D Visual Grounding models provides distinct advantages in overcoming these issues.

### 3 METHOD

As illustrated in Figure 2, GCML consists of three components. The Large Language Model is responsible for breaking down complex motion generation tasks into a combination of simpler ones, as well as generating control data for certain human joints in each subtask. The 3D Visual Grounding Model identifies navigable areas and locates the targets of interaction within the scene. The Motion Generation Model produces full-body motion sequences for each subtask based on the control

information provided by the other modules and integrates these sequences into a complete output motion.

This chapter begins by introducing the input and output formats of the task (Section 3.1). It then discusses how the LLM Planner and Subtask Executer convert the text description and scene point cloud into the control data necessary for the motion generation model (Section 3.2). Following this, we explain how the Motion Generation Model synthesizes the control data into a full motion sequence (Section 3.3)and provide further details on the 3D Visual Grounding Model used in our method (Section 3.4). Finally, we present the construction of the Complex Motion Evaluation Set(Section 3.5).

### 3.1 PROBLEM FORMULATION AND NOTATIONS

The generation of complex motions can be viewed as a recursive process, where complex motions are composed of simpler ones that follow the same input and output structure. Consequently, the task of generating complex motions can be divided into generating multiple simple ones and linking them together. Therefore, a method that can generate complex motions is inherently capable of generating simple motions as well.

Specifically, the input to our task consists of a user text prompt $T$ and a scene $S$, with the output being a human motion sequence $H$. The user prompt $T$ is a text description of the motion to be generated, while $S \in \mathbb{R}^{N \times 6}$ represents a point cloud of $N$ points, each with RGB color information. The output motion sequence $\{H_i\}_{i=1}^N$ is a series of human pose parameters over $N$ frames. We use the SMPL-X model (Pavlakos et al., 2019) to represent the human body's pose and shape. The SMPL-X body mesh $M \in \mathbb{R}^{10475 \times 3}$ is parameterized as $M = F(t, r, \beta, p)$, where $t \in \mathbb{R}^3$ is the global translation, $r \in \mathbb{R}^6$ is the continuous representation of global orientation, $\beta \in \mathbb{R}^{10}$ defines the body shape, and $p \in \mathbb{R}^{J \times 3}$ represents joint rotations in axis-angle format. $F$ is the differentiable linear blend skinning function. In most cases, we do not generate the full set of SMPL-X shape parameters directly; instead, we first generate the 3D world coordinates of 22 key body joints $P \in \mathbb{R}^{22 \times 3}$, which are then used to infer the complete SMPL-X parameters. The process can be summarized by the following formula:

$$T + S \longrightarrow \{P_i\}_{i=1}^{22} \longrightarrow M = F(t, r, \beta, p) \longrightarrow \{H_i\}_{i=1}^N \tag{1}$$

### 3.2 LLM PLANNER AND SUBTASK EXECUTOR

Similar to VoxPoser (Huang et al., 2023b), our method leverages a Language Model Program (LMP) to use a Large Language Model in generating human motion. Each LMP is responsible for a distinct function—such as breaking down tasks or invoking perception modules—and can call upon other LMPs as needed. The Large Language Model employed in our approach is GPT-4 (Achiam et al., 2023) from OpenAI.

Figure 3 provides an example of the generation process. First, we instruct the Large Language Model to learn from the provided code examples. For each LMP, we offer around 5-10 query samples along with their corresponding responses.

Following the sequence of calls between LMPs, the user's input motion description is first sent to the Task Planner, where complex or abstract tasks are broken down into simpler ones for the Subtask Executor. These tasks typically involve actions like walking to a location, interacting with an object, or performing a motion while standing still. The Subtask Executor creates human-centric motion descriptions based on the subtask, while object interaction information is provided by the 3D Visual Grounding model. Scene traversability data is directly obtained from the scene's point cloud. Spatial data is organized using a voxel map, with the Target Map and Avoidance Map values combined to produce a cost map $C \in R^{100 \times 100 \times 100}$, where lower values correspond to interaction targets and higher values indicate obstacles. Finally, using the following formula, we generate the trajectories of key human joints (such as the pelvis and hand), ensuring they avoid obstacles and reach the interaction target.

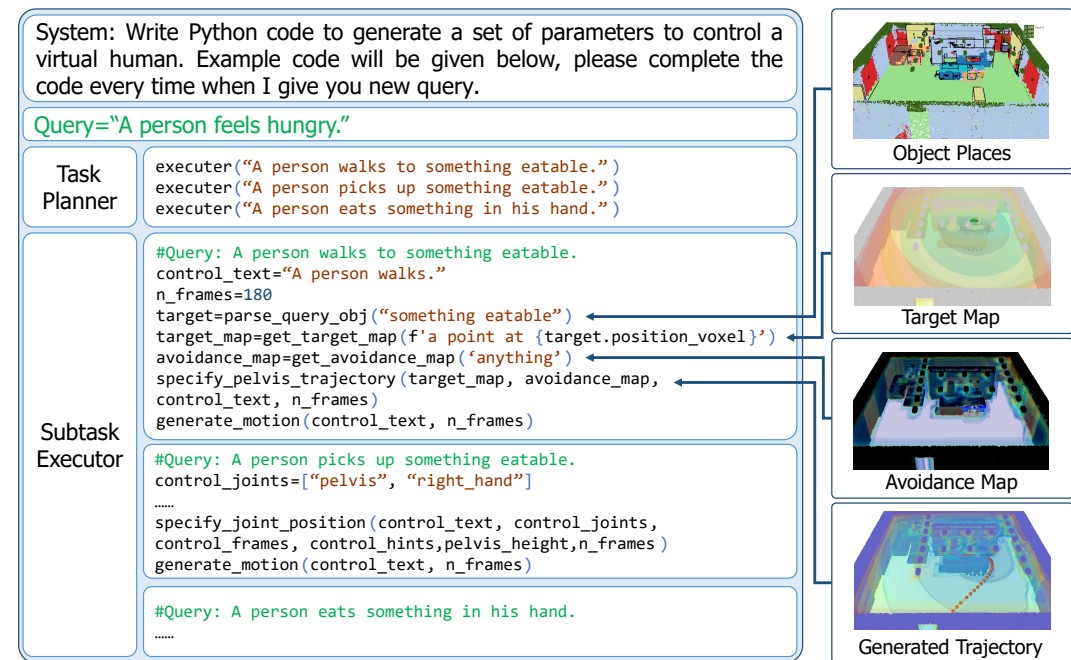

Figure 3: Example of how the Large Language Model generates human motion based on task descriptions and scene information. The Language Model Program (LMP) leverages provided examples to generate code that calls functions or other LMP instances from the user's query. This code then uses the perceived scene information to invoke the Motion Generation Model and produce complete motion frames.

$$p = \arg\min_{p_i} \sum_{i=1}^{n} \left[ C(p_i) - w_{\text{inertia}} \cdot \langle p_i - p_{i-1}, d \rangle + w_{z_{\text{penalty}}} \cdot |p_{i,z} - p_{i-1,z}| \right] \quad (2)$$

where $p$ represents the generated trajectory of human keypoints, $w_{\text{inertia}}$ is the inertia weight, which helps prevent the trajectory from getting stuck in local minima, $w_{z_{\text{penalty}}}$ is the z-axis offset weight, which ensures the human does not move over obstacles when controlling pelvis. As a result, we obtain control data for the human joints $h \in R^{N \times 22 \times 3}$, along with a human-centric motion description in text form. These data are then passed to the Motion Generation Model to produce a complete human motion sequence.

### 3.3 MOTION GENERATION MODEL

Our Motion Generation Model is built upon OmniControl (Xie et al., 2023), a diffusion-based generative model that conditions on both text and spatial keypoint positions. OmniControl was trained on the HumanML3D dataset and supports control over any joints at any time. We adapted it to generate motion sequences of arbitrary length.

In the previous steps, we obtained joint control data $h \in R^{N \times 22 \times 3}$. At each step of diffusion, we calculate the L2 distance between the generated joints and the control data $h$, using the gradient to guide the model in generating sub-motion sequences that align with both the action description $T$ and the spatial control signal $h$. To ensure smooth transitions between sub-sequences, we use the motion inbetweening method from MDM (Tevet et al., 2023). In each diffusion step, we replace the first frame of the generated motion with the last frame from the previous sequence. We also compute a transformation matrix based on the movement from the first to the last frame of the previously generated sequence. This matrix is applied to the subsequent sequences, merging the sub-task sequences into a coherent long motion sequence.

## 3.4 3D Visual Grounding Model

In the above generation process, we did not elaborate on how the positions of required objects were obtained from the scene. While many well-established methods exist for detecting and localizing 3D objects (Caesar et al., 2020); (Chen et al., 2020), in the scenarios described in this paper, we sometimes need to detect unconventional objects (e.g., "something eatable" in Figure 3). For this reason, we chose OpenScene (Peng et al., 2023) as our scene perception module due to its support for open vocabulary queries. Once OpenScene assigns object categories to each point in the scene's point cloud, we use the DBSCAN (Khan et al., 2014) clustering algorithm to filter out noise and pinpoint object instance locations. Additionally, OpenScene's ability to handle open-vocabulary queries allows us to detect object parts as well. For example, when interacting with a door, we can specifically locate the door handle rather than the entire door, which helps create more realistic motions.

## 3.5 Complex Motion Evaluation Set

In addition to controlling the behavior and style of the virtual human, we are also interested in understanding how they would respond to environmental changes and perform everyday tasks. The former requires the virtual human to simulate human thinking and react appropriately to external stimuli, while the latter involves decomposing actions into subtasks that are easier to execute. These complex actions are difficult to generate, even though they are quite common in everyday life. To evaluate model's ability to generate these intricate motions, we established the Complex Motion Evaluation Set.

We gathered 16 scenes from ScanNet (Dai et al., 2017) and Replica (Straub et al., 2019) as environments for virtual human activities, annotating these scenes with their offsets from the origin. We then provided the scene information and multi-view RGB images to a vision-language model (VLM). By asking, "What could an advanced virtual human do in this scene?" we generated a series of possible behaviors. After part-of-speech tagging, these behaviors were transformed into HSI descriptions that guide interactions between the virtual human and the environment. Compared to manually designing interactions, this automated approach saves labor and ensures that the generated descriptions are free from personal bias.

The generative model is tasked with producing complex human motions based on the scene mesh and the aforementioned HSI descriptions. Its performance is evaluated using relevant metrics and a human perceptual study. The HUMANISE dataset, Generalization Evaluation Set, and our newly introduced Complex Motion Evaluation Set all use scenes and text as conditional inputs, generating human motions that align with the semantic meanings of both the text and the scenes. Table 1 presents examples cases from each dataset. They show increasing difficulty levels and pose greater challenges for motion generation methods.

## 4 Experiments

We tested our method on the widely-used HUMANISE dataset for generating simple motions. For more complex motions, we evaluated the method on Generalization Evaluation Set, which is specifically designed to assess the generalization ability of models to unseen cases. Additionally, we tested our model's capacity to generate difficult yet common human motions in our newly introduced Complex Motion Evaluation Set.

## 4.1 Evaluation Metrics

**Generation Metrics:** When evaluating on the Humanise dataset, we followed the evaluation protocols of (Wang et al., 2022b) and (Zhang et al., 2020). Specifically, we used goal distance to measure grounding accuracy, contact to assess the realism of contact between the generated motion and the scene, and non-collision to quantify the proportion of actions that did not collide with the scene. In the afford-motion Generalization Evaluation Set and our Complex Motion Evaluation Set, we applied the same physical metrics but excluded goal distance, as these tasks did not have a clearly defined target object for interaction. Additionally, in alignment with afford-motion evaluation, we utilized the metrics proposed by (Guo et al., 2022a) to assess the quality of the generated motions.

Table 1: **Example cases of three datasets.**

| Evaluation Set | Scene | Control Text |
|---|---|---|
| HUMANISE | 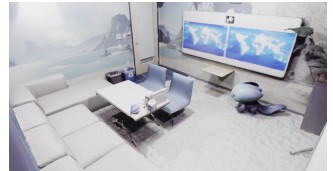 | 1. Walk to the desk.
2. Walk to the door.
3. Stand up from the chair.
4. Sit on the chair.
5. Lie down on the sofa. |
| Generalization Evaluation set | 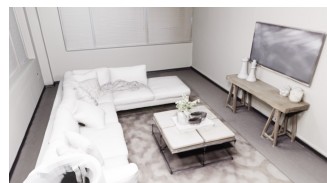 | 1. A person takes a rest on the sofa.
2. A person is shaking hands with someone.
3. A man jumps to the desk like a rabbit.
4. A man dances on the bed happily.
5. Someone sits on the edge of the bed. |
| Complex Motion Evaluation Set | | **Human Centric Motions:**
1. A tired person returns home.
2. A thief enters the room looking for something.
3. An earthquake is coming and a person feels it.
**Object Maneuver Motions:**
1. A person throws something into the dustbin.
2. A person moves the desk to the sofa.
3. A person arranges the room for a meeting. |

Table 2: **Experimental Results on HUMANISE dataset. Bold** indicates the best result.

| Method | Goal Distance↓ | Contact↑ | Non-collision↑ | Quality score↑ | Action score↑ |
|---|---|---|---|---|---|
| HUMANISE (Wang et al., 2022b) | 0.422 | 0.8406 | 0.9977 | 2.25 | 3.66 |
| Afford-motion (Wang et al., 2024) | 0.156 | **0.9568** | 0.9970 | 3.46 | **4.47** |
| Ours | **0.115** | 0.9526 | **0.9979** | **3.85** | 4.20 |

These include the Fréchet Inception Distance (FID) for evaluating the naturalness of the generated motions, R-Precision to gauge the alignment between the generated motions and the text prompts, and a diversity metric to measure the variability in the generated actions.

**Perceptual Study:** We also conducted a human perceptual study to assess the quality of the generated motions and their consistency with the corresponding text and scene. Participants rated the overall motion quality, including naturalness and collision levels, on a 1-5 scale, recorded as the quality score. They also rated how well the generated motions performed the actions specified in the text and interacted with target objects in the scene, recorded as the action score. Higher ratings indicated better alignment with the text and scene. We enlisted 20 evaluators to score 180 generated motion clips and recorded their average scores.

## 4.2 RESULTS ON HUMANISE DATASET

The HUMANISE dataset is considered the first widely adopted HSI dataset. It aligns motion sequences from the AMASS dataset with 3D scenes from ScanNet (Dai et al., 2017) and employs an automated annotation method to synthesize paired data rich in human-scene interaction information. However, HUMANISE is limited to simple actions like walking, sitting, and lying down, which constrains its applicability. Table 2 shows the quantitative results of the HUMANISE baseline, afford-motion, and our method for generating simple actions. Our method is primarily designed for generating complex motions. As a result, the task planner typically decomposes HUMANISE tasks into two steps: move to the interaction target and perform the interaction. Despite this, our method surpasses others on metrics such as goal distance, non-collision rate, and the quality score

Table 3: **Experimental Results on Afford-Motion's Generalization Evaluation Set.** "Real" indicates that these data are reference metrics from the HumanML3D test set. "→" indicates metrics that are better when closer to "Real" distribution.

| Method | FID↓ | R-precision (Top-3)↑ | Diversity → | Contact↑ | Non-colllision↑ | Quality score↑ | Action score↑ |
|---|---|---|---|---|---|---|---|
| Real | 0.000 | 0.875 | 9.442 | - | - | - | - |
| Afford-Motion | **7.887** | 0.478 | **7.935** | 0.7198 | **0.9983** | 2.06 | 2.63 |
| Ours | 8.215 | **0.687** | 7.677 | **0.9520** | 0.9972 | **4.03** | **4.22** |

Table 4: **Experimental Results on Our Complex Motion Evaluation Set.**

| Method | FID↓ | R-precision (Top-3)↑ | Diversity → | Contact↑ | Non-colllision↑ | Quality score↑ | Action score↑ |
|---|---|---|---|---|---|---|---|
| Real | 0.000 | 0.875 | 9.442 | - | - | - | - |
| Afford-Motion | 10.97 | 0.300 | 8.087 | 0.7277 | 0.9961 | 2.24 | 1.72 |
| Ours-text only | 43.75 | 0.251 | 3.754 | 0.6955 | 0.9915 | - | - |
| Ours-w/o planner | 12.57 | 0.362 | **8.219** | **0.9104** | 0.9973 | 2.09 | 2.41 |
| Ours | **9.31** | **0.365** | 7.987 | 0.8444 | **0.9979** | **3.15** | **3.17** |

based on human evaluations. Notably, our method excels in goal distance, demonstrating its ability to precisely guide the virtual human to the target and initiate the interaction.

### 4.3 RESULTS ON GENERALIZATION EVALUATION SET

Researchers in the HSI field have increasingly sought to generate more than just routine actions like walking or sitting. This set contains 16 scenes from ScanNet (Dai et al., 2017), PROX (Hassan et al., 2019), Replica (Straub et al., 2019), and Matterport3D (Chang et al., 2017), along with 80 carefully crafted HSI descriptions. These descriptions often specify multiple aspects of a single action, such as "a person jumps to the desk like a rabbit", which not only requires the virtual human to jump in a rabbit-like manner but also defines the target location as the desk. Such descriptions are rarely found in existing training datasets, posing significant challenges for motion generation methods in terms of generalization. Table 3 presents the performance metrics of afford-motion and our method on the Generalization Evaluation Set. While our method scores slightly lower than afford-motion on FID and Diversity, it significantly outperforms it on R-precision, indicating better adherence to the text and scene constraints. This is further corroborated by a larger gain in both quality score and action score.

### 4.4 RESULTS ON COMPLEX MOTION EVALUATION SET

Section 3.5 describes the details of our proposed Complex Motion Evaluation Set. We evaluated the performance of Afford-Motion and GCML on this evaluation set. As shown in Table 4, our method consistently outperforms afford-motion on all metrics in this new test set. In many cases, afford-motion only generates irrelevant or meaningless actions, while our method can break down complex action descriptions into sequences of subtasks and execute them accordingly. Our method performs particularly well in R-precision and contact metrics, indicating that it closely follows textual instructions and exhibits rich interaction with the environment. The perceptual study shows that our method achieves promising results in both overall generation quality and adherence to conditions. Notably, our approach allows precise control of action duration based on the text, supporting the generation of interaction sequences lasting up to several tens of seconds.

### 4.5 ABLATION STUDY

As shown in Table 4, firstly, we directly pass the text prompts to the motion generation model without using the whole pipeline we proposed. In this case, the generated results were scene agnostic, leading to a significant drop in physical metrics compared to others. Furthermore, a more critical

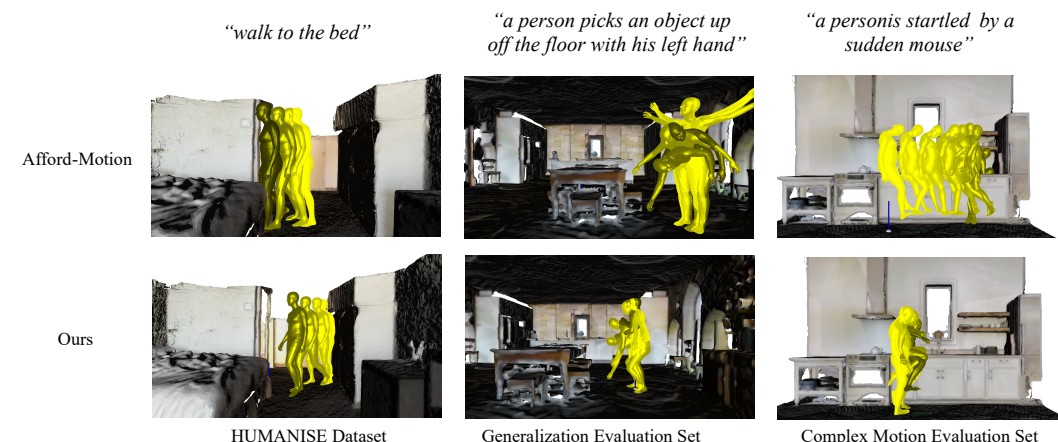

Figure 4: Comparison of the qualitative results between the Afford-Motion and our method on various datasets. Brighter actions precede darker actions.

issue arose: the motion generation model struggled to comprehend the actual intent behind complex user instructions, resulting in highly distorted human motions in most cases.

Next, we investigated the effect of task decomposition on motion generation outcomes. Here, we skipped the planner stage and directly provided the executor with the text prompts and scene data to produce control commands and spatial information for the motion generation model. The results, shown in the fourth row of Table 4, reveal that the LLM planner plays a crucial role in enhancing motion quality. Without the planner, the LLM Executor generates spatial control data that is challenging for the Motion Generation Model to interpret and follow, thereby diminishing the overall quality of the generated motions.

## 4.6 QUALITATIVE RESULTS

Figure 4 illustrates the visualization results comparing our method with Afford-Motion across three datasets. The left column shows the generation results for simple actions on the HUMANISE dataset, where both methods produce satisfactory results. However, our method allows for the specification of the character's initial pose. The middle column presents the outcomes from the Generalization Evaluation Set. In this example, the user prompt indicates that the target should be grasped with the left hand, but Afford-Motion overlooks this instruction during the generation process. The rightmost column presents the generation results from our Complex Motion Evaluation Set. When text prompts do not explicitly specify the desired action, Afford-Motion generates irrelevant or distorted motions, while our method can infer the implicit motion directives. Here, our approach establishes a connection between actions like "avoiding with a lifted foot" and "a sudden mouse", enabling the generation of coherent character motions based on abstract prompts.

## 5 CONCLUSION

This paper introduces GCML, a novel method for generating complex human motions guided by textual descriptions within a scene. By utilizing a Large Language Model for task decomposition and subtask execution, and a 3D Visual Grounding Model for scene perception, our method produces complete complex motion frames. We validated the effectiveness of our method across multiple datasets, with experimental results showing that our approach performs well in generating simple human motions. Moreover, on our newly introduced test set for complex human motion generation, our method consistently outperformed existing methods across all evaluation metrics.

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
