# OpenReview forum: "GCML: Grounding Complex Motions using Large Language Model in 3D Scenes"
_ICLR.cc/2025/Conference — ICLR 2025 Conference Withdrawn Submission_

### Official Review · Reviewer_juCk · 2024-10-27

**Soundness:** 3
**Presentation:** 3
**Contribution:** 3
**Rating:** 5
**Confidence:** 4

**Summary:**

This paper proposes an approach to generate complex motions using a Large Language Model based on input consisting of a language goal and a complex scene. By incorporating a task planner using an LLM, the proposed approach can decompose the complex action sequences into several simple actions and then solve these simple action generations separately. By combining these simple action sequences, the approach can achieve diverse complex tasks involving full-body motions.

**Strengths:**

1: This paper addresses an important research question. I agree that generating complex full-body motions in various realistic scenarios is crucial for both the computer graphics and robotics research communities.

2: I like the idea of breaking down complex motion generations into several simpler problems using a task planner with large language models.

3: The paper is well-written, with high-quality and easy-to-understand figures.

4: The authors compare the proposed approach to several baselines through extensive experiments.

**Weaknesses:**

1: My first concern is the limited details provided in this paper. For example, there is no information about the prompts used for the large language model and vision language model. I would expect to see these details, at least in the supplemental material.

2: This paper does not discuss its limitations. Could your proposed approach be applied to real-world robots? Would it be robust to sensor noise and ambiguous language goals? Though answering these questions would offer more insights, I would encourage the authors to thoroughly investigate the limitations of their paper.

3: This paper also does not have a discussion of failure cases. Under what conditions would your proposed approach fail? In Table 2, your approach sometimes performs worse than the bassline (Afford-motion). What analysis can explain this result?

4: This paper misses some important related works on generating task planners with large language models and 3D visual grounding, such as [1,2].

References:

[1]: Y. Huang, C. Agia, J. Wu, T. Hermans, and J. Bohg. Points2Plans: From Point Clouds to Long-Horizon Plans with Composable Relational Dynamics, ArXiv, 2024.

[2]: K. Lin, C. Agia, T. Migimatsu, M. Pavone, and J. Bohg. Text2motion: From natural language instructions to feasible plans. Autonomous Robots, 47(8):1345–1365, 2023.

**Questions:**

1: what’s the difference between your proposed approach to [1][2]?

2: what are the limitations and failure cases of this paper?

---

### Official Review · Reviewer_aDUw · 2024-11-01

**Soundness:** 2
**Presentation:** 2
**Contribution:** 2
**Rating:** 3
**Confidence:** 5

**Summary:**

This paper aims to generate human motions in 3D scenes from natural language prompts. The language prompt is first decomposed into a sequence of simple atomic actions using GPT-4, and then each simple action is processed by the subtask executor to get the joint trajectories. Finally, a pretrained motion generation model from OmniControl (Xie et al., 2023) yields the final human motion conditioned on the decomposed action description and joint trajectories. The authors conducted experiments to show the proposed methods outperforms two baseline methods.

**Strengths:**

1. The proposed method is training-free and can directly be applied to given 3D scenes. It leverages GPT-4 for task decomposition, a pretrained OpenScene (Peng et al., 2023) model for object grounding, and a pretrained motion generation model OmniControl (Xie et al., 2023). All modules are readily available for immediate use.

2. The subtask executor considers the interaction between human and scenes, encouraging the human to reach the goal location and avoid colliding with obstacles using the target map and avoidance map.

3. Experiments show the proposed method outperforms two baseline methods in scene-text conditioned motion generation.

**Weaknesses:**

1. The presented results can not support the central claim of generating human-scene interactions, such as mopping floor(L40), brushing teeth, and watering plants (L123). These interaction examples are not presented in the submission. According to the presented results in the first supplementary video, there is no real interaction between the human and scene objects. In the presented example of washing dishes, the person does not really have contact with the dishes and just randomly waves hands in the air.

2. The generated motion quality is far from satisfactory. There exists a lot of human-scene penetrations in the presented video results, e.g., the sequence labelled as 'sit on the toilet'. Foot skating and jittering artifacts are obvious in all non-walking sequences. The results in the Complex Motion Evaluation Set even show weird, twisted bodies. The presented motion quality is far from being useful for real applications. I recommend the authors to aim for motion quality at least on par with TRUMANS (Jiang et al., 2024), Object Motion Guided Human Motion Synthesis (Li et al., 2023), and Human-Object Interaction from Human-Level Instructions (Wu et al., 2024).

3. Many important technical details are missing, especially for the subtask executor.  The missing information include: the prompts used for the task planner; how the initial human location in the scene is determined; what are the provided code examples to GPT for the Language Model Program (LMP); how is the target map and avoidance map is built; how the N-frame 22 joints trajectory in L306 is obtained from LMP and how the minimization in equation 2 is solved (I also have the question whether the output is a single joint trajectory as visualized in generated trajectory in Figure 3 or full body 22 joints trajectory as stated in L306).

4. With the limited presented information, the planner and subtask task executor are very similar to the method proposed in VoxPoser (Huang et al., 2023b), with a LLM-based decomposition planner, a vision language model for scene grounding and output python programs to build voxel value maps, and trajectory synthesis given the voxel value maps. Further clarifications about the distinction between the proposed method and VoxPoser are needed.

5. Although the subtask executor takes target and obstacle into consideration, the subsequent motion generation by OmniControl is scene-agnostic, which is a source for artifacts like scene penetration.

6. The visualization view in the video results is not informative enough. In the first video, most human bodies are occluded by the furniture, hiding the skating and jittering artifacts. The top-down view of the other videos also has scene or self-occlusion problems, I would suggest adding one more side-view visualization.

**Questions:**

1. Why is the cost map set a resolution of 100x100x100? This resolution may be sufficient for the tabletop object grasping scenario in VoxPoser (Huang et al., 2023b). However, indoor rooms typically have much larger scales, and a resolution of 100x100x100 can result in a too coarse voxelization that can not accurately represent the environment, especially for fine-grained object interactions. This coarseness could potentially contribute to the human-scene penetrations observed in the video results.

2. If the output in L306 is full body 22 joints trajectory as stated, I would appreciate visualization of this intermediate result and how different it is from the final generation of OmniControl.

---

### Official Review · Reviewer_4FAy · 2024-11-04

**Soundness:** 3
**Presentation:** 2
**Contribution:** 2
**Rating:** 5
**Confidence:** 3

**Summary:**

The paper introduces GCML, a framework designed to generate complex human motions in 3D scenes based on textual descriptions and scene context. The method is motivated by two key challenges in Human-Scene Interaction (HSI): the lack of diverse, high-quality paired datasets for complex actions and the limitations of existing models that primarily generate simple motions. GCML leverages a Large Language Model (LLM) to decompose complex tasks into simpler subtasks and uses a 3D Visual Grounding Model to identify interaction targets and obstacles within the scene. It then synthesizes full-body motion sequences that align with both the user's input and the scene's semantics. The paper's main contributions include the introduction of a new task and evaluation set for complex motion generation, outperforming existing methods in generating intricate, realistic motions. GCML demonstrates competitive performance on simple tasks and significantly excels on its proposed Complex Motion Evaluation Set.

**Strengths:**

- The LLM-based approach is sensible, which decomposes complex motion tasks into simpler ones and makes the whole task much more manageable.
- The method can take pretty ambiguous prompts like “a person feels hungry”, and generate a sequence of plausible motions, which is impressive.
- The proposed Complex Motion Evalution set demonstrates the advantage of the proposed method, and the dataset itself can be a good addition to advance research in this area.

**Weaknesses:**

- One main weakness is the novelty of the visual grounding part & motion generation parts of the framework, which is similar to [1] published at CVPR 2024. [1] also VLMs to ground target objects and generation motion based on it. That said, the LLM decomposition part still has its novelty, although subtask planning using LLMs is quite common.
- The generated motion has sudden jitter (e.g., 00:18-00:25 in the video), which is undesirable for real-world applications.
- The writing of the paper also needs improvement. Eq 2 is not well explained. What is d? And how is this objective optimized?

[1] Cen, Zhi, et al. "Generating Human Motion in 3D Scenes from Text Descriptions." *Proceedings of the IEEE/CVF Conference on Computer Vision and Pattern Recognition*. 2024.

**Questions:**

- The top down angle in the visual results makes it difficult to see the motion quality. It would also be nice to provide more visual examples showcasing the capability of the system.
- Are the generated subtask programs by LLM in Fig. 3 fully directly used to call the functions? E.g., avoidance_map, specify_joint_position, generate_motion. Would there be any errors or bugs in LLM’s generated programs? If so, how does the system handle them?

---

### Official Review · Reviewer_sp7L · 2024-11-04

**Soundness:** 3
**Presentation:** 2
**Contribution:** 3
**Rating:** 6
**Confidence:** 4

**Summary:**

The paper introduces GCML (Grounding Complex Motions using a Large Language Model), a framework for generating human interactions from textual and scene inputs. It combines technologies like GPT-4, OpenScene, and OmniControl to create an automated system for synthesizing long-term, complex human motions. A new evaluation set demonstrates the method's performance compared to existing approaches.

**Strengths:**

The framework's ability to generate complex, long-term human motions from scene and textual inputs could significantly benefit industries such as animation, gaming, etc.
The integration of LLMs and a 3D Visual Grounding Model automates the process of long-term human-scene interaction, potentially saving human efforts.

**Weaknesses:**

Several key related works should be discussed, including "Synthesizing Long-Term 3D Human Motion and Interaction in 3D" from CVPR 2021, which decomposes long-term human-scene interaction synthesis into subtasks of body generation and motion in-betweening. Also, "GOAL: Generating 4D Whole-Body Motion for Hand-Object Grasping" from CVPR 2022. It deals with whole-body motion synthesis involving hand-object interactions which I think is not solved very well in this paper. (could be a limitation) and "Language Models as Zero-Shot Planners: Extracting Actionable Knowledge for Embodied Agents" in ICML 2022, which shares similar concepts and outcomes even though it doesn’t directly generate human motion.

The quality of the generated motions remain unnatural, particularly at the junctions of sub-motion clips, which are noticeably disjointed. Could the authors consider using or referencing more state-of-the-art motion in-betweening methods, such as those discussed in "Flexible Motion In-betweening with Diffusion Models" in SIGGRAPH ASIA 2024, to enhance the naturalness of the generated motions?

There are issues with the notation used in the paper, such as the inconsistent use of the symbol 'N' in Lines 236 and 237 to represent both 'N points' and 'N frames', which should be distinctively defined to avoid confusion.

**Questions:**

I am interested in the generation time for a sequence and how time is distributed across the modules. If the process proves quick, it could be a valuable tool for artists in their creative workflows.

---

### Note · Authors · 2024-11-27

**Comment:**

Our work is not satisfactory enough, it still needs to be polished.

**Withdrawal Confirmation:**

I have read and agree with the venue's withdrawal policy on behalf of myself and my co-authors.